# Brain leptin reduces liver lipids by increasing hepatic triglyceride secretion and lowering lipogenesis

Martina Theresa Hackl [1], Clemens Fürnsinn[1], Christina Maria Schuh [1], Martin Krssak [1,2,3], Fabrizia Carli[4], Sara Guerra[4,5], Angelika Freudenthaler[1], Sabina Baumgartner-Parzer[1], Thomas H. Helbich [6], Anton Luger[1], Maximilian Zeyda [7], Amalia Gastaldelli [4,5], Christoph Buettner[8] & Thomas Scherer [1]

Hepatic steatosis develops when lipid influx and production exceed the liver's ability to utilize/export triglycerides. Obesity promotes steatosis and is characterized by leptin resistance. A role of leptin in hepatic lipid handling is highlighted by the observation that recombinant leptin reverses steatosis of hypoleptinemic patients with lipodystrophy by an unknown mechanism. Since leptin mainly functions via CNS signaling, we here examine in rats whether leptin regulates hepatic lipid flux via the brain in a series of stereotaxic infusion experiments. We demonstrate that brain leptin protects from steatosis by promoting hepatic triglyceride export and decreasing de novo lipogenesis independently of caloric intake. Leptin's anti-steatotic effects are generated in the dorsal vagal complex, require hepatic vagal innervation, and are preserved in high-fat-diet-fed rats when the blood brain barrier is bypassed. Thus, CNS leptin protects from ectopic lipid accumulation via a brain-vagus-liver axis and may be a therapeutic strategy to ameliorate obesity-related steatosis.

[1] Department of Endocrinology and Metabolism, Department of Medicine III, Medical University of Vienna, Spitalgasse 23, 1090 Vienna, Austria. [2] Department of Biomedical Imaging and Image-Guided Therapy, High-Field MR Center, Medical University of Vienna, Spitalgasse 23, 1090 Vienna, Austria. [3] Christian Doppler Laboratory for Clinical Molecular MR Imaging, MOLIMA, Waehringer Guertel 18-20, 1090 Vienna, Austria. [4] Institute of Clinical Physiology, National Research Council, Via G. Moruzzi 1, 56124 Pisa, Italy. [5] Institute of Life Sciences, Sant'Anna School of Advanced Studies, Via Santa Cecilia 3, 56127 Pisa, Italy. [6] Department of Biomedical Imaging and Image-Guided Therapy, Division of Molecular and Gender Imaging, Medical University of Vienna, Spitalgasse 23, 1090 Vienna, Austria. [7] Department of Pediatrics and Adolescent Medicine, Medical University of Vienna, Spitalgasse 23, 1090 Vienna, Austria. [8] Departments of Medicine and Neuroscience, and Diabetes, Obesity and Metabolism Institute (DOMI), Icahn School of Medicine at Mt Sinai, One Gustave L. Levy Pl, New York, NY 10029, USA. Correspondence and requests for materials should be addressed to T.S. (email: thomas.scherer@meduniwien.ac.at)

Fatty liver disease is associated with diabetes and obesity and represents the hepatic manifestation of the metabolic syndrome[1]. While the full pathophysiology of non-alcoholic fatty liver disease (NAFLD) remains incompletely understood, steatosis is commonly believed to be a key initiating event. Hepatic lipids accumulate when the sum of lipid influx and production exceeds the rate, at which the liver utilizes or exports triglycerides (TGs). Since TGs are exported after being packed into very-low-density lipoproteins (VLDL), the main carriers of TGs in plasma, proper production and secretion of hepatic VLDL particles are a critical prerequisite for the prevention of steatosis and NAFLD. The physiological importance of this process is illustrated in that both genetic ablation[2,3] and pharmacological inhibition[4] of microsomal triglyceride transfer protein (MTP), the rate-limiting enzyme for VLDL assembly, lead to hepatic steatosis. Apolipoprotein B (ApoB) is a major structural component of VLDL particles and its absence, be it due to genetic defects[5] or pharmacological intervention with antisense oligonucleotides[6], likewise results in hepatic lipid accumulation. Hence, VLDL export to shuttle TGs into adipose tissue represents a key disposal mechanism of excess lipids by the liver, thereby avoiding excessive ectopic lipid deposition and lipotoxicity.

Severe hepatic steatosis is also a hallmark of generalized lipodystrophy, a syndrome characterized by a lack of adipose tissue[7]. Leptin, an adipokine secreted principally by adipocytes, is dramatically reduced in patients with lipodystrophy. Notably, replacement therapy with recombinant leptin potently reverses hepatic steatosis, even though adipose tissue mass does not recover[8,9]. Besides hypoleptinemia also leptin resistance, a condition commonly observed in the obese state, seems to be a major driver of hepatic triglyceride accumulation and hence steatosis[10]. The main mechanism accounting for the anti-steatotic effect of leptin action has not been clearly delineated, but some believe that leptin replacement improves hepatic steatosis merely by ameliorating hyperphagia[8,11]. This view has recently been challenged by evidence that leptin treatment improves hepatic lipid content chiefly independently of reduced calorie consumption in lipodystrophic patients[12]. If so, stimulation of hepatic VLDL export represents a potential anti-steatotic mechanism of leptin action, which has not yet been thoroughly investigated.

In mild steatosis, the liver can compensate to some extent for excessive energy supply by accelerating hepatic TG export[13], but VLDL secretion plateaus with the progression to NAFLD, which suggests that endogenous signals promoting hepatic lipid storage progressively outweigh those driving hepatic lipid export. VLDL secretion can be regulated by the brain and the autonomic nervous system[14–18]. We have recently shown that such a signal can be induced by insulin action in the brain. Insulin delivery directly to the central nervous system (CNS) boosts hepatic lipid export and protects from NAFLD[19], albeit this opposes the direct insulin effects via receptors on hepatocytes, which result in inhibition of hepatic VLDL secretion[19–21]. Accordingly, systemic hyperinsulinemia[22,23] and long-acting insulin analogs with hepatic specificity[24] lead to hepatic steatosis. These findings suggest that an imbalance in the brain-liver interorgan crosstalk could play an important role in hepatic TG retention and steatosis leading to the development of NAFLD.

Since leptin predominantly acts via signaling through specific receptors expressed in the brain[25,26] and since leptin replacement in lipodystrophy counteracts hepatic steatosis, we hypothesized that leptin, like insulin, could boost liver lipid export and protect from NAFLD via receptors expressed in the CNS. This idea is further supported in that systemic leptin injections in ob/ob mice, which lack a functional leptin gene, lower hepatic lipid content[27] and db/db mice, which lack a functional leptin receptor gene, suffer from enlarged steatotic livers[28]. Peripherally injected leptin commonly fails to induce clinically significant anorexic and glucose lowering effects in obese or diabetic humans[29,30] due to their leptin resistance. Leptin resistance has been attributed in part to a limited capacity for leptin transport across the blood–brain-barrier (BBB), so that the leptin concentration in the brain fails to rise proportionally with the increased circulating plasma levels associated with obesity[31,32]. However, leptin's anorexic effects can be rescued at least partially in obese rodents by direct leptin infusion into a cerebral ventricle, which bypasses the limited leptin transport across the BBB[33,34]. A possible brain-dependent rather than direct mode of leptin action on hepatic steatosis might also explain the normal hepatic lipid content observed in mice with a liver-specific knock-out of the leptin receptor[28]. Thus, direct delivery of leptin into the CNS likely circumvents potential leptin transporter defects that gradually develop in obesity.

In the present study, we report a series of acute and chronic studies in male Sprague Dawley (SD) rats demonstrating that CNS leptin signaling both promotes hepatic TG export and decreases de novo lipogenesis in the liver. We show that the resulting reduction in hepatic lipid content occurs independently of differences in body weight and calorie intake, and we provide evidence that leptin action on liver lipid metabolism is preserved in obesity when leptin is delivered directly into the CNS. These results identify leptin as a circulating peptide that potently counteracts hepatic steatosis via action on the brain, and suggest that impaired leptin transport across the BBB represents a potential key contributor to obesity-related development of NAFLD.

## Results

**ICV leptin activates STAT3 in the MBH, PVN and DVC**. To explore the effects of brain leptin signaling on hepatic TG secretion, male SD rats were fitted with stereotaxic infusion cannulae targeting the 3rd ventricle of the brain. After recovery from the surgery, fasted and freely moving rats were intracerebroventricularly (ICV) infused with leptin according to the experimental protocol (Fig. 1a). An ICV leptin infusion activated the leptin receptor signaling cascade as assessed by phosphorylation of the signal transducer and activator of transcription (STAT) 3 in protein lysates of punch biopsies in several periventricular brain regions, i.e. the mediobasal hypothalamus (MBH), the paraventricular nucleus of the hypothalamus (PVN) and the dorsal vagal complex (DVC). Brain leptin increased STAT3 phosphorylation at Tyr705 in all three examined brain regions 2 to 3-fold. (Supplementary Figure 1a and b). Plasma leptin levels were similar between groups during the 4-hr acute ICV leptin infusion (Supplementary Table 1), suggesting comparable direct effects of circulating leptin on the liver. This notion was further supported by the finding that no change in liver STAT3 or STAT5 phosphorylation was observed after 4 h of ICV leptin infusion (Supplementary Figure 1c, and d).

**ICV leptin boosts liver TG secretion and reduces steatosis**. We next studied hepatic TG secretion after injecting a tyloxapol bolus via a jugular venous catheter. Tyloxapol blocks VLDL hydrolysis and thereby the clearance of VLDL particles so that all newly synthesized VLDLs accrue in plasma. Since chylomicrons produced in the gut are not a significant source of plasma TGs in animals fasted for >4 h, the progressive accumulation of TGs in plasma allows calculation of the hepatic VLDL secretion rate[15]. An acute ICV leptin infusion increased hepatic triglyceride export by about 30% (Fig. 1b, c) in comparison to a vehicle infusion (artificial cerebrospinal fluid; ACSF). In agreement with an increased rate of VLDL particle production ApoB100 in plasma

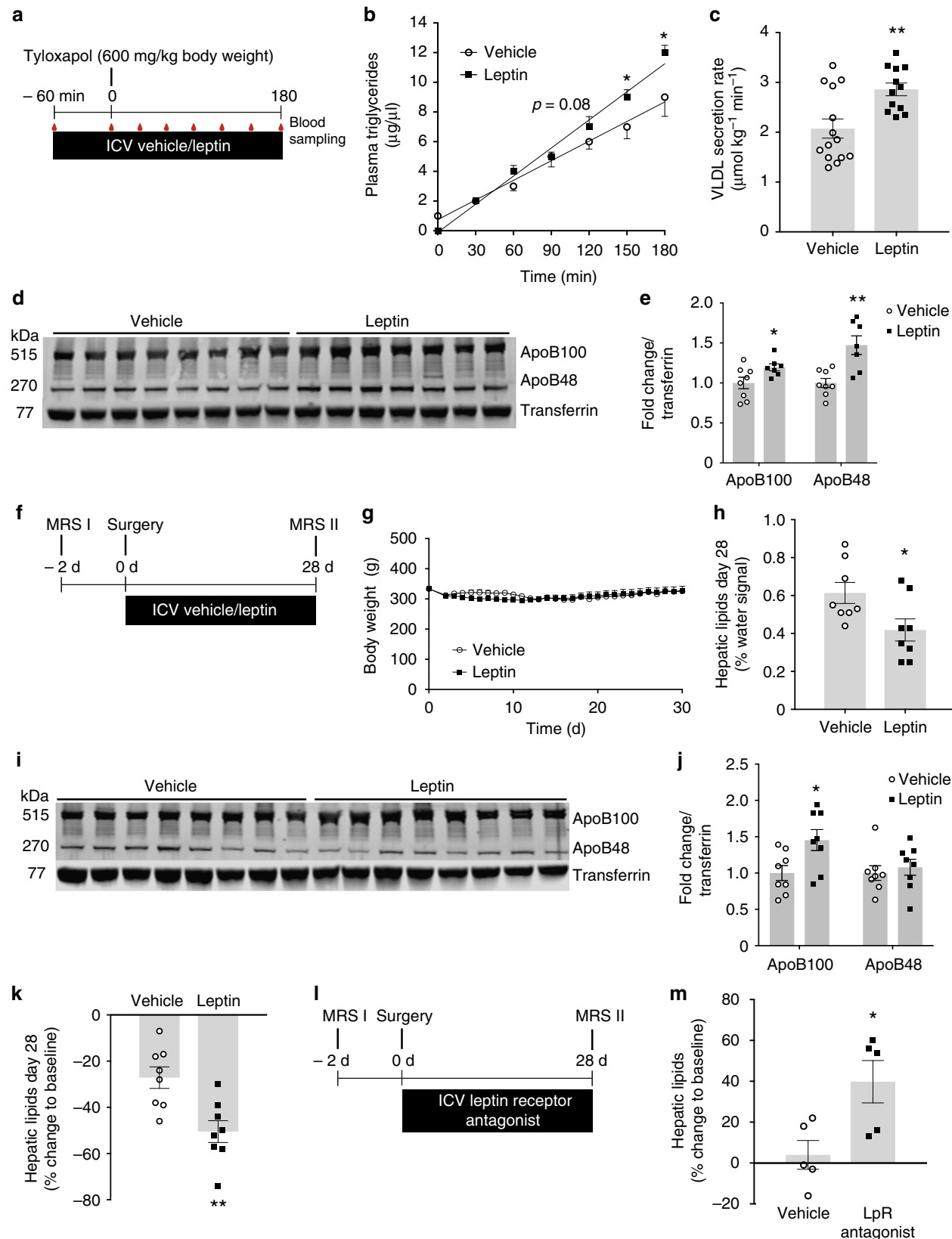

was increased by 20%. Rat hepatocytes also produce ApoB48, which was also upregulated after an acute ICV leptin infusion (Fig. 1d, e). The leptin-induced mobilization of TGs occurred independently of differences in plasma levels of glucose (Supplementary Figure 2a), insulin, free fatty acids, free glycerol and ketone bodies (Supplementary Table 1), and the rise in the VLDL secretion rate during short term (4 h) leptin infusion was not sufficient to cause an immediate detectable change in the final

**Fig. 1** Brain leptin signaling increases liver TG secretion and reduces hepatic steatosis. **a** Protocol for acute ICV leptin infusion experiments. **b** Plasma TG accumulation in ICV leptin/vehicle-infused rats after a tyloxapol bolus injection (Leptin: 1 μg/h; $n \geq 12$ per group). **c** VLDL secretion rate calculated from the slopes depicted in Fig. 1b. **d** Western blot of ApoB100 and ApoB48 in plasma samples at timepoint 180 min from acute ICV leptin/vehicle infusion experiments. **e** Quantification of the Western blot analysis from Fig. 1d ($n \geq 7$ per group). **f** Protocol for chronic ICV leptin/vehicle experiments. **g** Body weights and **h** hepatic lipid content assessed by $^1$H-MRS after 28 days of chronic ICV leptin/vehicle infusion (Leptin: 0.3 μg/day). **i** Western blot analysis of ApoB100 and ApoB48 from plasma after chronic leptin/vehicle infusion collected at the end of the experiment. **j** Quantification of the Western blot analysis in Fig. 1i ($n = 8$ per group). **k** Relative changes compared to baseline in hepatic lipid content after 28 days of chronic ICV leptin/vehicle infusion. **l** Protocol for chronic ICV leptin receptor antagonist experiments. **m** Relative changes in hepatic lipid content assessed by $^1$H-MRS during 28 days of blocking endogenous leptin signaling with an ICV infused peptide leptin receptor antagonist (Leptin receptor antagonist: 6 μg/day; $n = 5$ per group). All data are mean ± SEM; *$p < 0.05$; **$p < 0.01$; vs vehicle group by two-tailed Student's $t$ test; open circles: ICV vehicle; black squares: ICV leptin except for (**m**): ICV leptin receptor antagonist

hepatic lipid content (Supplementary Figure 2b). To assess, whether chronic stimulation of brain leptin signaling would affect liver lipid content in the longer time periods, we next infused ICV leptin or vehicle continuously for 4 weeks (protocol in Fig. 1f). Hepatic lipid content was measured non-invasively in anaesthetized rats using localized $^1$H-magnetic resonance spectroscopy ($^1$H-MRS) triggered to animal breathing, a method allowing consecutive non-invasive assessment of liver fat in vivo. The validity of the method was successfully confirmed by comparison to conventional TG measurements after Folch extraction (Supplementary Figure 2c). To exclude indirect effects via leptin induced reduction in food intake and body weight, control rats were calorically restricted to obtain matching weight curves in the two groups (Fig. 1g; Supplementary Figures 1d and e). In line with increased hepatic TG secretion observed in response to an acute leptin infusion, a one-month continuous ICV leptin infusion decreased hepatic lipid content by approximately 30% compared to the weight-matched vehicle-infused controls (Fig. 1h and Supplementary Figure 2f) and increased circulating ApoB100 levels (Fig. 1i, j). ApoB100 serves as an indirect measure of VLDL secretion, since each VLDL particle contains one ApoB100 molecule. Compared to baseline, leptin exposed animals lost twice as much liver fat as control rats (Fig. 1k), suggesting that only half of the leptin-induced reduction in liver fat can be attributed to weight loss caused by its anorexic action. Chronically leptin treated animals had to ingest slightly more food (Supplementary Figure 2d) in order to match the body weight of the controls, most likely because leptin increases energy expenditure[35]. Notably, the chronic leptin infusion not only improved hepatic TG content, but also reduced circulating free fatty acids, triglycerides, total ketone bodies, insulin and glucose levels, suggesting that metabolic health and insulin sensitivity were improved in a weight-independent manner (Supplementary Table 1). Conversely, blocking endogenous leptin signaling in the brain by infusing a leptin receptor antagonist (LpR antagonist; protocol in Fig. 1l) led to hepatic steatosis (Fig. 1m) without changes in body weight and food intake (Supplementary Figures 2g–i). The lack of orexigenic action of the LpR antagonist is likely due to the fact that we used a much lower dose of the LpR antagonist compared to other studies[36].

**Leptin effects on liver lipid metabolism are brain-mediated.** Beside the brain, hepatocytes also express leptin receptors[37], although these are predominantly short isoforms[38] that lack the intracellular domain required for the full range of signaling encompassed by the long form. To rule out that leakage of ICV infused leptin into the systemic circulation affects hepatic lipid content via direct action on the hepatocyte, we repeated the chronic infusion experiments with infusion of the leptin dose previously used in the ICV experiments via osmotic mini pumps implanted in the peritoneum (Fig. 2a). Intraperitoneal

(IP) leptin infusion had no effect on hepatic lipid content (Fig. 2b), food intake, body weight (Supplementary Figures 3a–c) and circulating glucose or lipids (Supplementary Table 2). In addition, we performed tyloxapol infusion studies (Fig. 2c) in fasted mice with a tamoxifen-inducible leptin receptor knockout (LpRΔPER) and compared them to age-matched tamoxifen-injected controls. In this strain of Rosa26 Cre-expressing mice tamoxifen deletes LpRs in peripheral tissues, but does not cross the BBB in sufficient amounts to cause a loss of LpRs in the CNS[26]. We found that such a specific reduction of peripheral LpRs (by ~70% in the liver; Supplementary Figure 3f) did not affect hepatic TG secretion (Fig. 2d, e), body weight or circulating glucose (Supplementary Figures 3d and e) indicating that the peripheral leptin receptors are not important in controlling hepatic VLDL secretion. The finding is consistent with a previous report that failed to detect a metabolic phenotype in these mice[26].

**CNS leptin suppresses hepatic de novo lipogenesis.** To better understand the molecular mechanism of the ICV leptin induced increased hepatic TG mobilization, we analyzed the protein expression of microsomal triglyceride transfer protein (MTP) in livers of 4 h ICV leptin-infused animals. MTP is the rate-limiting enzyme in VLDL assembly, which forms a heterodimer with protein disulfide isomerase (PDI), and consistent with the increased TG flux we observed a modest, but significant increase in its hepatic expression (Fig. 3a). Besides boosting TG secretion, CNS leptin may reduce liver lipids by inhibition of hepatic de novo lipogenesis. In support of this possibility, both acute and chronic leptin infusions reduced the protein expression as well as the enzyme activity of fatty acid synthase (FAS) in the liver. Acetyl-CoA carboxylase (ACC) was downregulated only after chronic brain leptin exposure. (Fig. 3a–f). Since FAS is a rate-limiting enzyme for de novo lipogenesis, these data suggest that brain leptin is also capable of reducing liver de novo lipogenesis. In agreement with this hypothesis, fatty acid profiles from liver tissue harvested after 4-weeks of ICV leptin infusion revealed a universal reduction in hepatic lipid species (Fig. 3g). Determination of individual lipid species allowed calculation of the hepatic stearoyl-CoA desaturase 1 (SCD1) index indicative of the activity of SCD1[39], the rate-limiting enzyme for the formation of monounsaturated fatty acids. The SCD1 index was markedly reduced by ICV leptin in the rats (Fig. 3h) consistent with previous studies reporting decreased hepatic SCD1 activity in ICV leptin-infused mice[40]. Congruent with the above findings that suggested a decrease in hepatic de novo lipogenesis, we observed a marked reduction in newly synthesized hepatic triacylglyceride (TAG) species after 2-weeks of ICV leptin infusion, as detected after the concomitant application of deuterated water ($D_2O$) and analysis through mass spectrometry to assess de novo lipogenesis in vivo (Fig. 3i, j). Plasma TAG enrichments significantly correlated with liver TAG enrichments across lipid species and

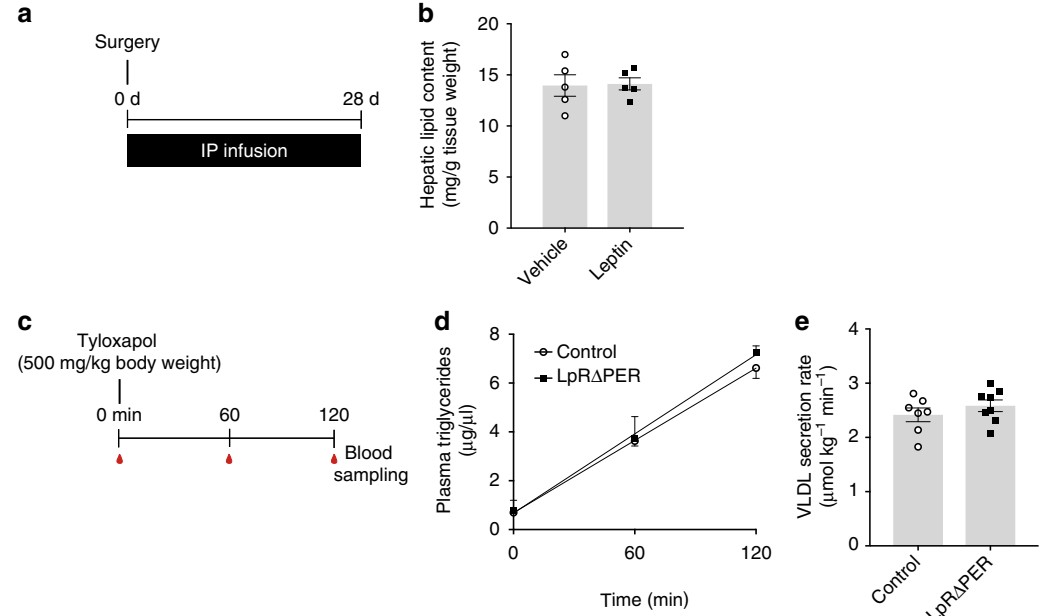

**Fig. 2** Leptin effects on hepatic lipid metabolism are centrally mediated. **a** Protocol for chronic IP leptin infusion in rats (Leptin: 0.3 µg/day; same dose used in the ICV experiments; Fig. 1f). **b** Hepatic lipid content after 28 days IP leptin/vehicle infusion ($n = 5$ per group). **c** Protocol for tyloxapol infusion experiments in tamoxifen-inducible LpRΔPER and control mice. **d** Plasma TG accumulation after a tyloxapol bolus in 6 h-fasted LpRΔPER and controls ($\geq 7$ per group). **e** Hepatic VLDL secretion rate calculated from the slope in Fig. 2d. All data are mean ± SEM; no significant differences observed between leptin and control by two-tailed Student's *t* test; (**b**) open circles: IP vehicle; black squares: IP leptin; (**d**, **e**) open circles: controls; black squares: tamoxifen-inducible leptin receptor knockout (LpRΔPER) mice

treatment groups suggesting that the plasma TAGs mirror hepatic lipid production (Supplementary Figure 2m). Total liver TAG content was reduced and we also detected a marked decrease in hepatic diacylglycerols (DAG) (Fig. 3k, l). These findings were independent from changes in body weight; and after 2 weeks of ICV leptin infusion cumulative food intake compared to weight-matched controls was similar (Supplementary Figures 2j–l). Thus, brain leptin reduces hepatic TG content not only by increasing hepatic VLDL secretion, but also by suppressing hepatic de novo lipogenesis.

**Hepatic vagal innervation is required to reduce steatosis.** Since there is evidence that the CNS regulates hepatic lipid metabolism via both the sympathetic and the parasympathetic nervous system[17,41], we next asked which branch of the autonomic nervous system conveys the leptin receptor signal from the brain to the liver. We performed both a liver-specific sympathectomy and a hepatic branch vagotomy and tested whether the effects of chronic ICV leptin were preserved (protocol depicted in Fig. 4a). Since microsurgical sympathectomy of the liver has some reproducibility issues even in experienced hands and might be complicated due to the trauma to the adjacent biliary and vascular structures[42], we used a bolus injection into the portal vein of 6-hydroxydopamine (6-OHDA), a neurotoxin that specifically targets sympathetic neurons and fibers. Similar studies performed previously in cats demonstrated a high first-pass effect for 6-OHDA[43]. Based on dose-finding studies (Supplementary Figures 4a), we selected a dose that reliably reduced hepatic norepinephrine levels by 90% with minimal effects at extrahepatic sites (Supplementary Figures 4b and c). However, selective hepatic sympathectomy did not affect ICV leptin's ability to lower hepatic lipid content (Fig. 4b) or circulating TGs (Supplementary Table 3). These results were obtained from groups of rats with comparable body weight due to food restriction of the control group (Supplementary Figure 4d–f). Contrary to the lack of

effects of a liver-specific sympathectomy, hepatic branch vagotomy completely abrogated the effects of ICV leptin on hepatic lipid content (Fig. 4c; Supplementary Figures 4g–i), suggesting that the vagus nerve conveys the leptin signal from the brain to the liver. We next examined whether this effect could also be observed in the acute setting (protocol in Fig. 4d). In line with the chronic infusion experiments, ICV leptin lost its ability to augment hepatic TG secretion (Fig. 4e, f) and reduce liver FAS activity (Supplementary Figure 4m) in rats subjected to hepatic vagotomy. Circulating levels of glucose, insulin and leptin were not affected (Supplementary Figure 4j and Supplementary Table 3), the latter confirming no significant leptin leakage from the brain into the peripheral circulation.

**Leptin receptors in the DVC modulate liver lipid metabolism.** We next examined which brain region mediates the anti-steatotic effect of brain leptin. Although the MBH is known to be involved in CNS–leptin-mediated effects on food intake, glucose, and lipid metabolism[44,45], an acute stereotaxic leptin infusion directly into the MBH using the same protocol depicted in Fig. 1a did not affect hepatic VLDL secretion (Fig. 4g, h). This is consistent with our previous findings on the role of CNS insulin[19], suggesting that the MBH is not a key anatomical region in the CNS to participating in the control of hepatic TG secretion. Since the efferent vagal motor neurons originate in the DVC, a brain region that could be involved in the regulation of CNS regulation of VLDL secretion[17], we next examined stereotaxic bilateral infusion of leptin directly into the DVC in tyloxapol infusion studies. Indeed, a DVC leptin infusion reproduced the effects of an ICV infusion into the 3rd ventricle (Fig. 4i, j), suggesting that stimulation of leptin receptors in the DVC trigger vagal signals that modulate hepatic lipid metabolism. Systemic leptin and glucose levels were unaffected in both, DVC and MBH infusion experiments (Supplementary Figures 4k and l; Supplementary Table 4).

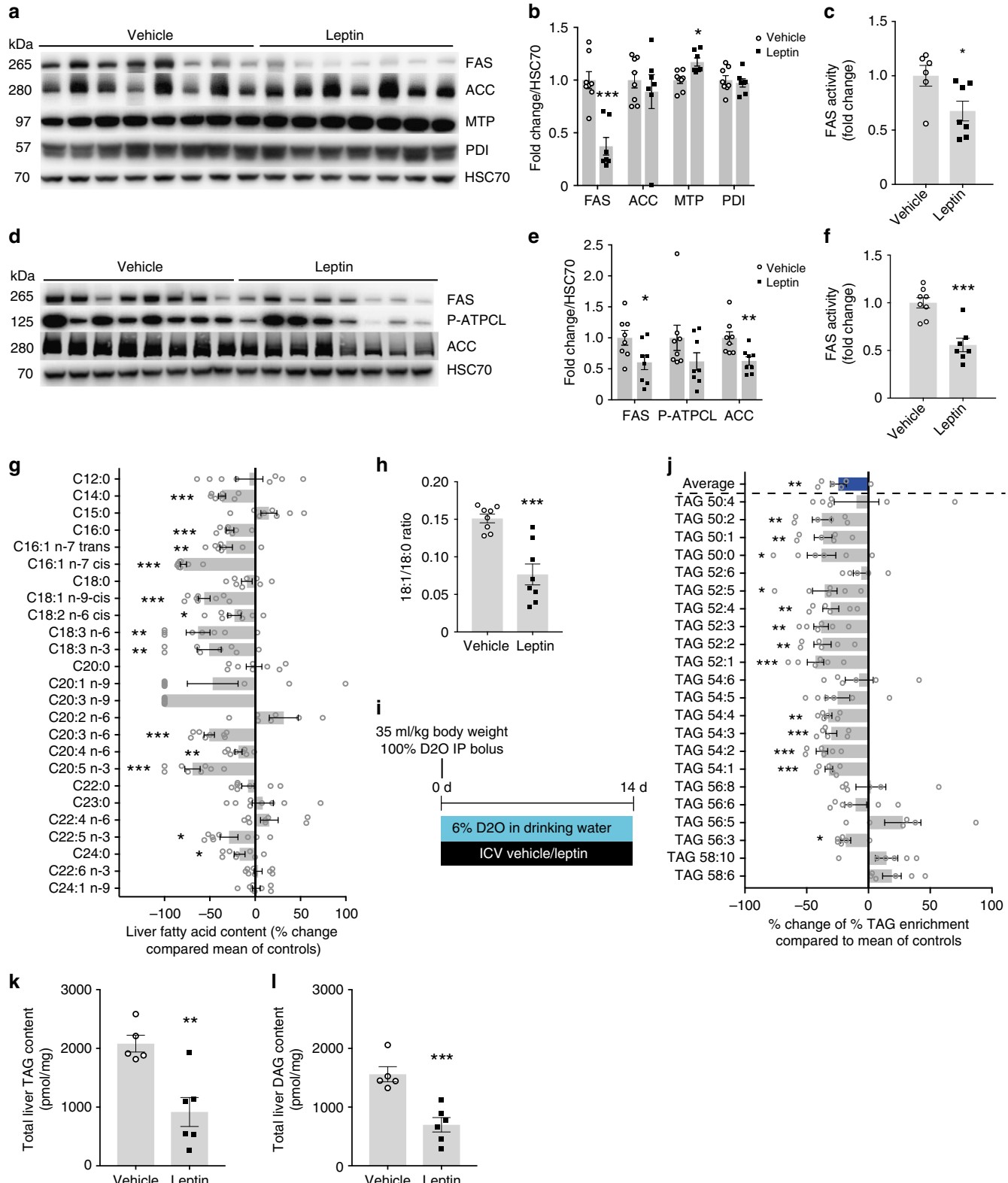

**ICV leptin has preserved anti-steatotic effects in obesity.** Having demonstrated that brain leptin signaling reduces hepatic lipid content under chow diet in metabolically healthy rats, we next examined these effects in metabolically challenged rats. Obesity commonly leads to leptin resistance and hyperleptinemia, which is associated with a failure of peripherally administered

leptin to improve metabolic control[46]. At least in rodents, however, ICV leptin administration partially restores the metabolic effects of CNS leptin signaling indicating that reduced BBB transport of leptin contributes to the leptin resistance[33]. We fed male SD rats with a 60% high-fat diet (HFD) for 4 weeks before and during chronic ICV leptin infusion (protocol depicted in

**Fig. 3** CNS leptin suppresses hepatic de novo lipogenesis. **a** Western blots for FAS, ACC, MTP and PDI of livers from acute ICV leptin infusion experiments (Fig. 1a). **b** Quantification of the Western blot analyses from Fig. 3a ($n \geq 7$ per group). **c** Liver FAS activity after acute ICV leptin/vehicle infusion ($n \geq 6$ per group). **d** Western blot analyses from liver tissue lysates of chronic leptin/vehicle infusion experiments (Fig. 1f). **e** Quantification of the Western blot analyses in Fig. 3d. **f** Liver FAS activity after chronic leptin/vehicle infusion ($n = 8$ per group). **g** Fatty acid profiles from liver tissues harvested on day 28 after chronic leptin or vehicle infusion ($n = 8$ per group). **h** SCD1 activity index calculated by the ratio of 18:1 (oleic acid) / 18:0 (stearic acid) from the fatty acid profiles in Fig. 3g. **i** Protocol for chronic 2-week infusion experiments with 6% $D_2O$ in drinking water. **j** De novo lipogenesis (DNL) triacylglyceride (TAG) species measured in liver tissues harvested after 2 weeks of chronic ICV leptin or vehicle infusion ($n \geq 5$ per group). In Fig. 3g and j data are depicted as % change vs vehicle-infused animals. **k, l** Hepatic TAG (k) and DAG (l) content ($n \geq 5$ per group). All other data are mean ± SEM; *$p < 0.05$; **$p < 0.01$; ***$p < 0.001$ vs vehicle group by two-tailed Student's $t$ test; open circles: ICV vehicle; black squares: ICV leptin

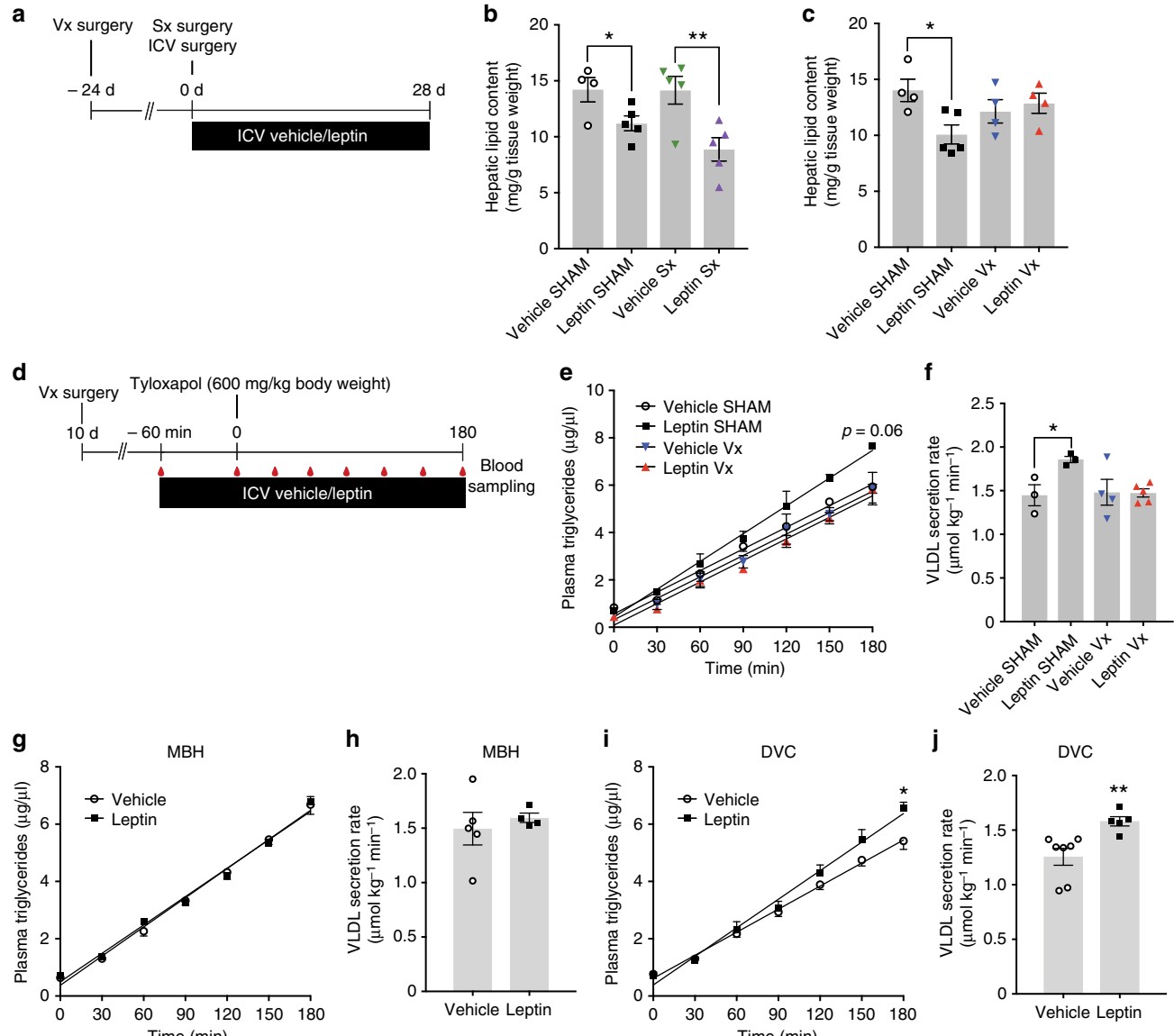

**Fig. 4** ICV leptin requires intact vagal innervation to regulate hepatic lipid metabolism. **a** Study timeline for chronic ICV leptin/vehicle infusion studies after selective hepatic sympathectomy (Sx) or vagotomy (Vx). **b, c** Hepatic lipid content after selective hepatic sympathectomy ($n \geq 4$ per group) (**b**) and liver vagotomy ($n \geq 4$ per group) (**c**) after 28 days of continuous ICV leptin/vehicle infusion (Leptin: 0.3 µg/day). **d** Study timeline and protocol for tyloxapol infusion studies in rats with hepatic vagotomy that received a 4 h ICV leptin/vehicle infusion. **e** Progressive accumulation of plasma TGs after an IV tyloxapol bolus ($n \geq 3$ per group). **f** VLDL secretion rates calculated from the slopes depicted in Fig. 4e. **g** Plasma increase in TG levels following a tyloxapol bolus after a targeted 4 h leptin/vehicle infusion directly into the mediobasal hypothalamus (MBH; $n \geq 5$ per group). **h** VLDL secretion rates calculated from the slopes in Fig. 4g. **i** Plasma increase in TG levels following a tyloxapol bolus after a targeted 4 h leptin/vehicle infusion directly into the dorsal vagal complex (DVC). **j** VLDL secretion rates calculated from the slopes in Fig. 4i ($n \geq 3$ per group). All data are mean ± SEM; *$p < 0.05$; **$p < 0.01$; vs vehicle group if not otherwise indicated by brackets. n.s. not significant by two-tailed Student's $t$ test; open circles: vehicle; black squares: leptin; green triangles pointing down: ICV vehicle plus liver Sx; violet triangles pointing up: ICV leptin plus liver Sx; blue triangles pointing down: ICV vehicle plus liver Vx; red triangles pointing up: ICV leptin plus liver Vx

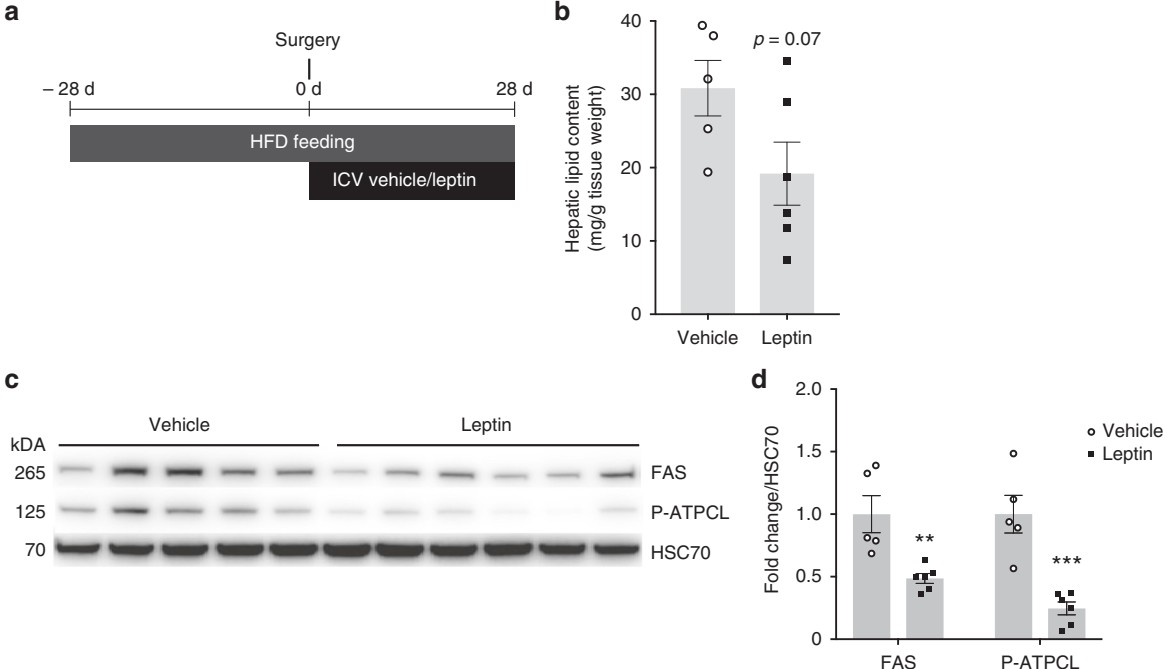

**Fig. 5** Brain leptin signaling has preserved anti-steatotic effects in obese animals. **a** Study timeline of HFD feeding combined with the chronic ICV leptin/vehicle infusion protocol. Note that we used a 3-time higher ICV leptin dose compared to previous experiments on regular chow (Fig. 1f). **b** Hepatic lipid content after 8 weeks of HFD feeding with ICV leptin/vehicle infusion during the last 4 weeks ($n \geq 5$ per group). **c** Western blot analyses of liver tissue lysates ($n \geq 5$ per group). **d** Quantification of the Western blot analyses in Fig. 5c. All data are mean ± SEM; **$p < 0.01$; ***$p < 0.001$; vs. vehicle by two-tailed Student's $t$ test; open circles: ICV vehicle; black squares: ICV leptin

Fig. 5a). Due to the higher fat mass that we assumed to cause leptin resistance, we increased the daily ICV leptin dose by 3-fold. Again, body weights of the vehicle infused control rats were matched by a restricted feeding protocol (Supplementary Figures 5a–c). Under these conditions, a weight-neutral reduction of hepatic lipid content with ICV leptin by 30% compared to controls and almost equivalent hepatic TG content compared to chow-fed controls (Supplementary Figure 2f) suggested that the anti-steatotic action of ICV leptin was largely preserved (Fig. 5b). As in previous experiments, the expression of key proteins involved in hepatic de novo lipogenesis was markedly suppressed by ICV leptin infusion (Fig. 5c, d). Notably, in chronically ICV leptin-infused HFD-fed rats leptin signaling, as assessed by STAT3 phosphorylation at Tyr705 in biopsies of the MBH and the DVC, is preserved (Supplementary Figure 5d and e). However, when acutely injected IP, leptin failed to induce STAT3 phosphorylation in the DVC and the MBH, respectively (Supplementary Figure 5f and g), which suggests that peripherally injected leptin is not adequately transported across the BBB in the obese state. Overall, these data suggest that ICV-delivered leptin activates STAT3 signaling in the CNS and retains its anti-steatotic effects even in the obese state.

## Discussion

Here we demonstrate that brain leptin signaling increases TG export and decreases de novo lipogenesis in the rat liver, which within weeks reduces hepatic lipid content independently of changes in body weight and food intake, and without any accompanying hypertriglyceridemia. Within the CNS, the DVC seems to be a key site mediating the effects of leptin on hepatic lipid metabolism, which depend on the parasympathetic innervation of the liver (Fig. 6). Our studies therefore indicate that leptin functions as a physiological signal that acts via a brain-

vagus-liver axis to protect the organism from ectopic lipid accumulation in the liver. In this context it is important to point out that, while there are considerable differences in sympathetic innervation patterns of the liver between rats and humans, cholinergic nerve fiber distribution shows very similar patterns in both species[47,48]. Thus, we speculate that this newly characterized regulatory pathway makes an important contribution to the amelioration of hepatic steatosis in subjects with lipodystrophy under leptin treatment, which until now has often been attributed solely to leptin induced anorexia and weight loss[8,11]. Because recent research claimed that hepatic steatosis is still improved through strict control of caloric intake in individuals with lipodystrophy[12], the present study provides a potential mechanism, through which leptin improves hepatic steatosis beyond its well-characterized effects on appetite and body weight. Since we documented this only for a rodent model, future studies are needed to evaluate whether this mechanism is also at work in humans. Studies in healthy humans and patients with lipodystrophy examining hepatic VLDL secretion will also provide key evidence for a potential utility of recombinant leptin as a novel drug for treating obesity-related NAFLD, for which no clinically effective therapy is yet available.

Although peripheral leptin receptors exist, the metabolic effects of leptin seem to be predominantly mediated via the brain[25,28]. Nevertheless, there are reports that leptin directly modulates lipid metabolism in isolated rat livers, albeit with somewhat contradictory outcomes ranging from leptin-induced accumulation to depletion of hepatic lipids[49,50]. Ultimately, we cannot entirely exclude that direct effects of leptin in the liver contribute to leptin action following systemic administration, but in our studies we did not detect chronic effects on hepatic lipid content when leptin was infused systemically (IP) (Fig. 2a, b). When administered at equal doses centrally (ICV), we saw a marked reduction in hepatic lipid content independent of differences in body weight or

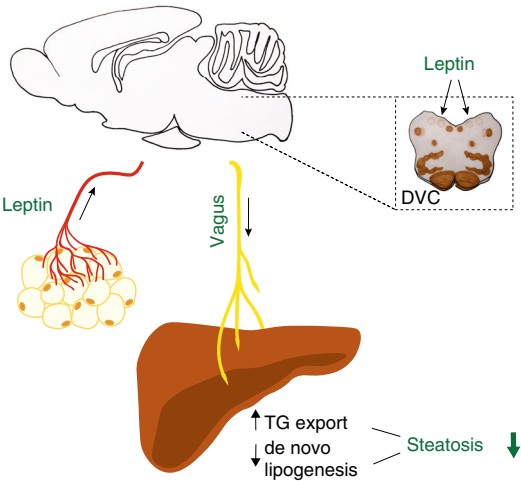

**Fig. 6** Proposed model of the role of CNS leptin in regulating hepatic lipid metabolism. Leptin secreted by white adipocytes protects the liver from ectopic lipid accumulation and lipotoxicity. Leptin reaches the brain by passing through the blood–brain barrier to increase TG secretion and reduce hepatic de novo lipogenesis via signaling in the dorsal vagal complex, where the efferent vagal motor neurons are located. These centrally mediated leptin effects require intact liver vagal innervation and are preserved under HFD-conditions when leptin is administered directly into the brain

diet (Figs. 1g and 5b). Together with a 20–30% increase in the VLDL secretion rate, which was triggered by both acute leptin infusion ICV or directly into the DVC (Figs. 1b and 4i), our results argue for a clinically important role of brain-dependent leptin action on hepatic lipid handling. In support of this conclusion, mouse models with normal LpR expression in the brain, but reduced LpR expression in the liver or in the whole periphery (as presented in our study) developed no apparent steatotic phenotype[26,28] and showed unchanged VLDL secretion rates during tyloxapol infusion studies (Fig. 2d). On the contrary, genetic models with disrupted brain leptin signaling such as in db/db mice and neuronal LpR knock-out mice, have enlarged steatotic livers[26,28]. Accordingly, blocking endogenous leptin signaling in the brain led to hepatic lipid accumulation in our study (Fig. 1m). Taken together, all these findings strongly argue for a predominantly centrally mediated leptin effect and at best, a minor contribution of liver LpRs in the regulation of hepatic lipid metabolism. Since we found that only a chronic leptin infusion into the cerebroventricular space led to measurable changes in hepatic lipid content, we believe that leptin, which is chiefly secreted in proportion to body fat stores, regulates hepatic lipid content rather long-term than in the minute to minute control of metabolism such as in the fasting to feeding transition. This notion is supported by the findings that leptin levels are only moderately modulated in humans by feeding, fasting or circadian rhythmicity[51,52].

In addition to lipid export and de novo lipogenesis, leptin has also been associated with increased β-oxidation in various tissues including the liver[53,54]. Although we have not directly measured hepatic β-oxidation, lack of brain leptin-induced changes in hepatic protein expression of peroxisome proliferator activated receptor (PPAR) α, which is regarded as an important regulator of leptin-mediated lipid oxidation[54], does not suggest a major role for hepatic β-oxidation in our experimental setup (Supplementary Figures 6a and b). Furthermore, the marked 50% decrease in circulating ketone bodies and the fact that ICV leptin infusion

does not change lipid-derived circulating long-chain acyl-carnitines, which are typically increased in situations of stimulated β-oxidation such as fasting and HFD feeding[55], also argue against an increase in hepatic lipid oxidation (Supplementary Table 1 and Supplementary Figure 6c and d). Others reported comparable results and showed a suppression in ketogenesis when leptin was infused systemically in a hyperglycemic model of insulin-deficient type 1 diabetes[56]. Our study demonstrates that chronic CNS leptin at very low concentrations (50x lower compared to the systemic infusion used in[56]) reduces ketogenesis also in the normoglycemic, normoinsulinemic state independent of differences in body weight, again highlighting the key role of the CNS in leptin action.

We further observed a reduction in circulating plasma leptin in rats subjected to chronic ICV leptin infusion, which occurred under both regular chow diet and HFD despite comparable body weights in treated and control animals (Supplementary Tables 1 and 5). This response was presumably secondary to specific loss of fat mass triggered by ICV leptin, that went clearly beyond of the reduction found in controls that were weight-matched by food restriction (ICV leptin vs vehicle: 78% reduction in epididimal fat mass under chow conditions; $p = 0.0002$ vs vehicle by a two-tailed Student's test and 44% reduction under HFD conditions; $p = 0.07$ vs. vehicle by a two-tailed Student's test; Supplementary Tables 1 and 5) as also observed by others[57,58]. These effects are likely due to central leptin's ability to activate lipolysis and suppress de novo lipogenesis in white adipose tissue[45,57]. Of note, the observation of systemic hypoleptinemia in chronically ICV leptin treated rats also adds to the evidence against a peripheral hormonal action of leptin.

Leptin resistance is associated with hepatic TG accumulation[10,59] and high circulating leptin levels, which are likely a reflection of increased leptin resistance, and correlate with NAFLD[60]. Leptin transport across the BBB is saturable[61]. Thus, despite increased systemic leptin levels in the obese, leptin resistant, state, relative hypoleptinemia in the CNS develops. Our findings in HFD-fed animals that ICV leptin has preserved effects on brain STAT3 signaling, hepatic lipid content and de novo lipogenesis provides evidence for the concept that leptin resistance can be partially overcome by direct CNS administration[33,34]. Since neuropeptides such as leptin can be delivered into the CNS via an intranasal route of administration, intranasal leptin treatment could be a possible future treatment strategy in NAFLD. Proof of concept experiments in rodents have already been conducted and delivered promising result, e.g., leptin´s anorexic effect was preserved in diet-induced obese rats receiving leptin intranasally[62]. In addition, it was demonstrated with radio-labeled leptin that systemic hyperleptinemia does not hamper the uptake of intranasally administered leptin into the brain[63]. Adding further support to this concept, we show here that intranasal administration of leptin acutely phosphorylates STAT3 in the DVC indicating rapid leptin uptake into the brain (Supplementary Figure 1e). Our studies support the idea that direct targeting of leptin signaling pathways in the CNS can circumvent a potential leptin transporter defect and, hence, could have therapeutic potential in the treatment of NAFLD. Based on these findings it seems promising to test if intranasal leptin overcomes leptin resistance in patients with obesity and type 2 diabetes. However, while intranasal leptin administration shows promising results in rodent studies a clinical application in humans has certain limitations: Relatively high doses have to be applied when peptide hormones are administered intranasally. Since recombinant leptin is expensive a human trial with intranasal leptin is currently not viable. In addition, variable absorption via the nasal mucosa may be an issue. However, possible approaches to circumvent some of these drawbacks could be to

engineer leptin analogs with improved BBB permeability features[64,65] or to co-infuse leptin with peptides or drugs that restore leptin sensitivity in the obese state[66].

In conclusion, we show that leptin signaling in the brain protects from ectopic lipid accumulation in the liver by stimulating TG secretion and reducing lipid production without inducing hypertriglyceridemia. Notably, these effects are mostly preserved in obese animals when leptin is administered directly into the 3rd ventricle. Therefore, stimulation of brain leptin signaling, could be a strategy to ameliorate the hepatic steatosis not only in lipodystrophy but also in conditions such as obesity and diabetes.

## Methods

**Animals and ethical approval.** All experiments were performed in approximately 10-week-old male Sprague Dawley (SD) rats fed either a standard chow diet (sniff R/M-H Alleinfutter, sniff Spezialdiaeten GmbH, Soest, Germany) or a high-fat diet (HFD; 60% of calories from fat; D12492, Research Diets, Brogaarden, Denmark). Rats were purchased from the Department of Laboratory Animal Studies and Genetics of the Medical University of Vienna, Austria or from Janvier Labs, France. All animals received tap water ad libitum and were maintained at room temperature in individual cages at a 12-h light/12-hour dark cycle. For all surgical procedures rats were initially anesthetized with a mixture of ketamine and xylazine and then kept under isoflurane inhalation anesthesia after intubation with a modified 14-gauge intravenous (IV) cannula (Venflon, BD, Switzerland). We complied with all relevant ethical regulations for animal testing and research and the experimental procedures were approved by the Austrian Federal Ministry of Science, Research, and Economy (BMWFW-66.009/0246-WF/V/3b/2015). Mouse experiments were approved by the International Animal Care and Use Committee of Mount Sinai School of Medicine, New York, NY (IACUC LA09-00174).

**Acute tyloxapol infusion studies.** Tyloxapol studies (protocol outlined in Figs. 1a and 4d) were performed as described elsewhere[19]. Briefly, rats were implanted with a 22-gauge single guide cannulae (PlasticsOne, Roanoke, VA) targeting the third ventricle (ICV; coordinates: 2.5 mm posterior from the bregma on the sagittal suture, 9 mm below the cortical surface) or a 26-gauge double cannula targeting either the dorsal vagal complex (DVC; coordinates: 13.3 mm posterior from the bregma, 0.6 mm from midline, 6.9 mm below the cortical surface) or the medio-basal hypothalamus (MBH; coordinates: 3.3 mm posterior from bregma, 0.4 mm from midline; 9.6 mm below the cortical surface). In addition, all rats received an indwelling intra-jugular venous catheter for blood sampling and injection. Prior to the infusion experiment, we allowed rats to recover from surgery in individual cages for 5–7 days. Food intake and body weight were monitored routinely and they were used only if returned to within 10% of their pre-surgical body weight.

The tyloxapol solution (Triton WR-1339, T0307, Sigma Aldrich, Germany) was prepared at a concentration of 80 mg/ml in PBS, pH 7.4. Animals were fasted for 5 to 6 h prior to the tyloxapol infusion experiment in order to exclude overestimation of hepatic VLDL flux caused by triglyceride (TG) appearance in plasma by chylomicron secretion from the gastrointestinal tract. Fasted rats received a continuous ICV, DVC or MBH infusion over 4 h via an infusion cannula with a 1 mm projection (PlasticsOne, Roanoke, VA) that was inserted into the pre-implanted guide cannula (time point −60 min to 180 min; ICV infusion rate 0.083 μl/min, DVC and MBH infusion rate 0.0055 μl/min/min/side) with either leptin (ICV 1 μg/h; DVC and MBH 0.05 μg/h; Protein Laboratories Rehovot Ltd., Israel) or artificial cerebrospinal fluid as the vehicle (ACSF, Tocris, UK). At time point 0 min rats were given an IV tyloxaypol bolus (600 mg/kg) via the pre-implanted venous catheter. During the whole experiment rats were conscious and allowed to move freely in their cage. Blood samples were taken via the venous catheter at time points −60, 0, 30, 60, 90, 120, 150 and 180 min for measurement of plasma TG. At the end of the infusion period rats were killed by a lethal IV injection of ketamine followed by decapitation. The VLDL secretion rate was calculated from the slope of the TG accumulated in plasma over time by linear regression analysis, under the assumptions that the average molecular weight of TGs is 884 g/mole and the average volume of distribution of VLDLs is 0.042 ml/g body weight.

**Chronic brain infusion.** The chronic brain leptin infusion studies were conducted in SD rats as outlined in Figs. 1f, l and 5a. Rats were surgically implanted with brain stereotaxic infusion cannulae targeting the third ventricle (ICV; coordinates: 2.5 mm posterior from the bregma on the sagittal suture, 10 mm below the cortical surface; PlasticsOne, Roanoke, VA). Cannulae were connected to an osmotic minipump (ALZET Model 2004, Cupertino, CA) positioned subcutaneously in the neck region allowing a continuous ICV infusion for 28 consecutive days with either leptin (0.3 μg/day for chow-fed animals, 0.9 μg/day for HFD-fed animals), a leptin receptor antagonist (LpR antagonist; 6 μg/day; Protein Laboratories Rehovot Ltd., Israel), or with vehicle (ACSF, Tocris, UK). After surgery, animals were single housed. Food intake and body weight were monitored daily and control rats in the ICV leptin infusion experiments were food restricted to maintain similar body

weight with the corresponding leptin-infused group. Food-restricted animals on chow received one portion of food daily in the afternoon. We frequently checked for the feeding pattern and found that they consumed their food portion in several meals throughout the day rather than showing any binge eating behavior. To further assure absence of binge eating in HFD food-restricted animals, these groups received their food in three portions evenly distributed over the day. Hepatic lipid content was assessed non-invasively by either ¹H-MRS two days prior to cannula implantation and 28 days post-surgery in briefly isoflurane-anesthetized rats or by modified Folch extraction in liver tissues post-mortem. Note, that ¹H-MRS was performed using MR-compatible ICV infusion kits and osmotic minipumps. The animals were killed 28 days post-surgery and liver tissue and EDTA treated plasma were harvested for further analysis.

**Liver ¹H- MRS.** Magnetic resonance spectroscopy was performed on a Biospec 9.4 T/30 cm MR system (Bruker Biospin, Ettlingen, GER) using an 86 mm ¹H tuned resonator for transmit and four channel radiofrequency-coil array for signal reception. Animals were anaesthetized by an isofluran/oxygen gas mixture in a prone position on an animal cradle supplied by the system manufacturer. Vital functions (ECG, breathing frequency, temperature) were monitored by an MR-compatible Small Animal Monitoring and Gating System (SA Instruments Inc., Stony Brook, NY, USA) and the body temperature was kept at 37 °C with water heated pads. Measurements of intrahepatic lipids were performed by a short echo time single voxel MRS sequence (Stimulated Echo Acquisition Mode, TE = 3 ms, TR = 5000 ms, NS = 32) without water signal suppression. The volume of interest of $6 \times 6 \times 6$ mm³ was placed within the liver parenchyma and data acquisition was synchronized with the breath movement of the animal. MRS data were processed off-line and signal intensities of water (W) and methylene group of lipids (L; CH2-1.25 ppm and CH-3 0.9 ppm) were used to quantify lipid content in the liver as intrahepatic lipid content = [L/(L + W)] and given in per cent.

**Chronic intraperitoneal leptin infusion study.** Animals were intraperitoneally implanted with osmotic minipumps (ALZET Model 2004, Cupertino, CA) prefilled with leptin or vehicle (ACSF, Tocris, UK; protocol in Fig. 2a). We infused the same dose as in the chronic ICV infusion experiments IP (0.3 μg/d) to rule out that leaking of ICV infused leptin into the periphery causes direct effects on the liver.

**Inducible peripheral LpR KO mouse (LpRΔPER mice).** Tamoxifen-inducible LpRΔPER (LpR^flox/flox Rosa26^Cre-ERT2/+) and control mice (LpR^flox/flox) were a gift from Yiying Zhang and Streamson Chua Jr. (both Columbia University, NY). The study (protocol outlined in Fig. 2c) was conducted 5 to 10 weeks after the tamoxifen induction (IP tamoxifen injections at a dose of 1 mg/day per mouse over 5 consecutive days) between 20 to 32 weeks of age. We achieved a 70% knockdown of the LpR in the liver as assessed by SYBR Green qPCR (Supplementary Figure 3f). The primer sequence for wild-type LpR was (forward) 5′-GGACTGAATTTC CAAAAGCCTG-3′ and (reverse) 5′-CATAGCTGCTGGGACCATC-3′ and (forward) 5′-GCATACAGGTCCTGGCATCT-3′ and (reverse) 5′-CCATCCAGCC ATTCAGTCTT-3′ for CYP A. Data were analyzed using the delta-delta-$C_t$ method ($2^{-\Delta\Delta Ct}$). After a 5–6 h fast a baseline blood sample was collected by tail vein sampling at time point 0 min. Tyloxapol (500 mg/kg body weight) was then injected via the tail vein. Mice were restrained for 2–3 min during the injection of tyloxapol, but were able to move freely in individual cages after that, except during the rapid collection of additional blood samples 1 and 2 h after the injection of tyloxapol. After the last sampling (time point 2 h) mice were anesthetized with isoflurane and killed by cervical dislocation.

**De novo lipogenesis in vivo analyses.** Rats were fitted with stereotaxic brain infusion cannulae targeting the 3rd ventricle that were connected to osmotic minipumps positioned subcutaneously in the neck region. Rats received either vehicle or leptin ICV over the course of 2 week. The detailed protocol is depicted in Fig. 3i. The surgical procedures were the same described above for the chronic infusion protocol. After the surgery all rats received a 35 ml/kg body weight $D_2O$ with 0.9% NaCl i.p. bolus injection. Rats then continued to receive 6% $D_2O$ in their drinking water. TAG and DAG species from liver tissue were extracted with a modified Folch method. Briefly, about 25 mg of liver tissues were homogenized with a tissue-lyser (2 cycles of 1 min at 25 Hz), and DAG and TAG species were extracted with 600 μl of chloroform:methanol (3:1) and 100 μl of water, with the addition of internal standards. Lipid classes were separated with UHPLC (1290 Infinitiy, Agilent, Santa Clara, CA) equipped with ZORBAX Eclipse Plus C18 2.1 × 100 mm 1.8 μm column and deuterium enrichment in the respective TAG species was measured with mass spectrometry (6540 QTOF, Agilent, Santa Clara, CA). TAGs and DAGs were quantified using the internal standards: TAG (15:0/15:0/15:0) and DAG (17:0/17:0), respectively (Larodan, Solna, SE). Deuterium enrichment in the TAG species was measured by mass spectrometry quantifying the $m + 1/m + 0$ area ratio corrected for background; de novo lipogenesis was quantified by dividing the deuterium enrichment in the TAG species by the number of hydrogens exchanged, normalized by precursor water enrichment (6%)[67]. To get a broader view on hepatic de novo lipogenesis, we analyze the incorporation of deuterium (% enrichment) in each distinct TAG class. For plasma TAG analysis 20 μl plasma was deproteinized with 200 μl of cold methanol and surnatant was

injected into the instrument. For the plasma/liver TAG correlation analysis in Supplementary Figure 6a only plasma TAG classes that contributed >0.1% to the total TAG peak area were included.

**Vagal hepatic branch vagotomy.** Vagotomy of the hepatic branch of the ventral subdiaphragmatic vagal trunk or sham surgery were performed as described by others[68]. In short, a small incision on the ventral midline was made. The gastro-hepatic ligament was severed to remove connective tissue between the liver and the esophagus for exposure of the descending ventral esophagus and the ventral sub-diaphragmatic vagal trunk. The hepatic branch was visualized using 0.1% toluidine blue dissolved in saline and transected. To minimize the possibility of nerve recovery, disconnected nerve endings were dabbed with a 10% phenol solution in 70% ethanol. The abdominal wall and the skin were closed with sutures. For the sham surgery, the same procedure was applied except for the transection of the vagal nerve branch and phenol treatment. Acute ICV leptin/vehicle infusion in the tyloxapol studies were performed after a 10-day recovery period. Studies with chronic ICV infusion of leptin/vehicle by osmotic minipumps were initiated after 4-weeks of recovery from the surgical intervention.

**Liver-specific sympathectomy.** Liver-specific sympathectomy was performed in SD rats by a bolus injection into the portal vein of 6-hydroxydopamine (25 mg/kg; 6-OHDA), a neurotoxin specifically targeting sympathetic nerve fibers, or vehicle (0.1% ascorbic acid in saline; sham controls). Note that the appropriate dose of 6-OHDA was determined in a separate dose-finding study, where the injected doses ranged from 10 to 50 mg/kg body weight. For the intraportal 6-OHDA injection, a midline incision along the linea alba was made. The colon was gently arranged onto a sterile cotton gauze soaked in saline until the mesenteric vessels were clearly visible. The mesenteric vein was cannulated with a portex fine tubing (ID 0.58 × OD 0.96 mm) and the tip of the tubing was advanced to the liver hilus into the portal vein. 6-OHDA was then slowly injected over 2 min and the tubing was removed sealing the puncture site with tissue glue. The colon was returned to the abdominal cavity, and the muscle and skin incisions were closed with sutures. Immediately after the sympathectomy the rats were implanted with a brain infu-sion kit and osmotic minipumps to initiate the ICV chronic infusion studies as described above. Due to a significant first pass effect, the liver was the primary target organ of the pharmacological sympathectomy, while other tissues were not significantly affected (Supplementary Figures 4a and b), which was validated by measuring tissue norepinephrine levels by ELISA (LDN, Nordhorn, Germany).

**Leptin signaling studies.** Brain signaling studies described in Supplementary Figures 1a and b were performed in freely moving SD rats that have been fasted for 5 h and were pre-implanted with ICV cannulas as described above. After inserting an infusion cannula with a 1 mm projection into the guide cannula conscious rats were given a bolus injection of leptin (10 μg in 10 μl, Protein Laboratories Rehovot Ltd., Israel) or vehicle (ACSF, Tocris, UK) and sacrificed 30 min after the bolus using isoflurane-anesthesia and decapitation. Tissue samples from the arcuate nucleus, paraventricular nucleus, and dorsal vagal complex were collected using a punch biopsy kit after freezing the respective sagittal brain sections in liquid nitrogen. Brain biopsies of the DVC and the MBH of HFD-fed rats (D12492, Research Diets, Brogaarden, Denmark, over 8-weeks) were harvested 60 min after an IP injection of leptin (1 mg/kg body weight). In vivo leptin signaling was assessed by analyzing the phosphorylation state of STAT3 in protein lysates from these biopsies by Western blotting using phospho-specific antibodies.

**Intranasal leptin signaling study.** The intranasal leptin infusion study in Sup-plementary Figure 1e was performed in rats under ketamine and xylazine anes-thesia. Either 1 mg/kg leptin (Protein Laboratories Rehovot Ltd., Israel) dissolved in saline or control solution (saline) were instilled equally into both nostrils of the animals fasted for 5 h. After 30 min rats were sacrificed and punch biopsies were collected from the DVC. The phosphorylation state of STAT 3 was assessed using Western blot with a phospho-specific antibody.

**Analytic procedures.** All tissue samples were immediately snap frozen in liquid nitrogen after harvesting. Blood samples were collected in EDTA tubes. Blood glucose was measured with a commercially available glucometer (One Tou-ch®Ultra, Milpitas, CA). Plasma triglycerides and free glycerol were measured with a colorimetric kit by Sigma Aldrich, Germany. Plasma leptin and insulin were measured by ELISA (leptin: Phoenix Pharmaceuticals, Burlingame, CA; insulin: Mercodia, Uppsala, Sweden). Plasma glucose levels (Human-Glucose liquiUV, Wiesbaden, Germany), ketone bodies and plasma free fatty acid levels (both Wako Chemicals, Neuss, Germany) were measured with colorimetric assays. Hepatic TGs were extracted with a modified chloroform-methanol Folch extraction[19]. Livers (100 mg) were homogenized in a 3:1 chloroform-methanol mixture with the Pre-cellys apparatus (Precellys24, Bertin Technologies SAS; Montigny-le-Bretonneux, France), then incubated in glass tubes for 4 h at room temperature. After adding 0.9% saline samples were vortexed and centrifuged at 2000 rpm for 10 min. The lower organic phase was transferred to another glass tube and evaporated under nitrogen. Samples were saponified in a 3 M potassium hydroxide solution in 65% ethanol for 60 min at 70 °C in a water bath. TGs were measured with the above-

mentioned Sigma Aldrich kit. Tissue norepinephrine levels were measured with an ELISA kit (LDN, Nordhorn, Germany).

**Plasma fatty acid profiles.** Liver fatty acid profiles were analyzed using gas chromatography as previously described[19]. In brief, 10 mg liver tissue was homo-genized in chloroform-methanol (3:1) and after addition of 10 μg internal standard (heptadecanoic acid) lipids were extracted, separated by centrifugation (3000 × g; 10 min at 4 °C) and the lower phase subjected to evaporation under nitrogen. Fatty acid methyl esters were prepared by adding a (5:1) methanol-toluene mixture and acetyl chloride and heating to 100 °C (1 h). After cooling down to room tem-perature, addition of 6% sodium carbonate and centrifugation (3000 × g, 5 min. 4 °C), the upper layer (300 μl) was analyzed by gas chromatography using a 6890 N/5973 N GC-MSD system (Agilent, Santa Clara, CA) equipped with a PTV-injector and a DB-23 column (30 m, 0.25 mm ID, film thickness 0.15 mm, Agilent J&W), using helium as carrier gas. Peak identification and quantification were performed by comparison of integrated peak areas to standard chromatograms. All calculations are based on fatty acid methyl ester values.

**Plasma long-chain acyl-carnitines.** Lipid-derived long-chain acyl-carnitines were determined from plasma by flow injection analysis after derivatization by n-butylation on an Acquity TQD detector (Waters, Milford, MA). Deuterated internal standards for semiquantitative analysis as well as all other reagents were obtained from Recipe Chemicals (Munich, Germany). Procedures, transitions and calculations were used according to the manufacturer´s (Recipe) instructions.

**FAS activity assay.** Fatty acid synthase (FAS) activity in liver tissue samples was measured spectrophotometrically as described in ref. [69]. Briefly, liver tissue (150 mg) was homogenized in ice-cold buffer (pH 7.0) containing 0.25 M sucrose, 1 mM DTT, 1 mM EDTA, and a protease inhibitor cocktail (Roche, Nutley, NJ). The homogenate was centrifuged for 10 min at 20,000 × g at 4 °C to collect the fat-free infranatant. The infranatant was ultracentrifuged for 1 h at 100,000 g at 4 °C. The remaining infranatant was mixed 1:2 with 500 mM potassium phosphate buffer (pH 7.4) containing 0.5 mM DTT and incubated at 37 °C for at least 20 min for maximal activity of the sample. Additionally, 200 μl of 500 mM potassium phosphate buffer (pH 7.0) containing 0.1 mM EDTA, 1 mM β-mercaptoethanol and 5 μl of freshly prepared NADPH solution (10 mM NADPH in 100 mM potassium phosphate buffer pH 7.0) were added to a 96-well plate and incubated at 37 °C for at least 10 min. A concentration of 100 μl of the activated sample were added to each well and the reaction was started by adding either 10 μl substrate solution (5 mM Acetyl-CoA and 5 mM Malonyl-CoA in a ratio of 3:4 dissolved in 100 mM potassium phosphate buffer pH 7.0) or 10 μl blank solution (dH₂O and 5 mM malonyl-CoA in a ratio of 3:4 dissolved in 100 mM potassium phosphate buffer pH 7.0). Absorbance was measured for 20 min every minute at 340 nm and the slope was calculated. FAS enzyme activity (mU per mg protein) was defined as 1 nmole NADPH consumed per min/mg. To calculate the final activity an extinction coefficient of 6220 per M/cm was used.

**Western blot analysis.** Liver and brain tissue samples were homogenized with 40 mM ß-glycerophosphate, 2 mM sodium orthovanadate, 20 mM MOPS, 30 mM sodium fluoride, 10 mM sodium pyrophosphate, 2 mM EGTA, 5 mM EDTA, pH 7.4 and complete™ Protease Inhibitor Cocktail (Roche; Nutley, NJ) in Precellys tubes containing ceramic beads (Precellys24, Bertin Technologies SAS; Montigny-le-Bretonneux, France) and put on ice immediately after. The lysate was then transferred to autoclaved Eppendorf tubes and centrifuged at 13,000 r.p.m. for 20 min at 4 °C. Protein concentrations were measured with the Pierce BCA Protein Assay Kit (#23225, ThermoFisher Scientific Inc., Waltham, MA). Protein extracts were separated on 4–12% NuPAGE gels (Invitrogen, Carlsbad, CA) and blotted onto either PVDF-P membranes for chemiluminescence detection or PVDF-FL membranes for infrared detection (both Immobilon; Merck-Millipore; Burlington, MA). Membranes were blocked at room temperature for 1 h in either Odyssey LI-COR Blocking Buffer (LI-COR Biosciences, Lincoln, NE) 1:2 diluted in TBS or 5% nonfat dry milk powder (w/v) (#9999; Cell Signaling; Danvers, MA) in 1× TBS-T and incubated in primary antibodies overnight at 4 °C. Primary antibodies against ACC (Cat. 3676, Lot #6) phospho-ATPCL Ser455 (Cat. 4331, Lot #2), PDI (Cat. 3501, Lot #3), GAPDH (Cat. 5174, Lot #4), STAT3 total (Cat. 4904, Lot #7) and phospho-Stat3 Tyr705 (Cat. 9145, Lot #26) (all Cell Signaling Technology); FAS (Cat. 610962, Lot #3266988), MTP (Cat. 612022, Lot #3266983) (both BD Bioscience); PPARα (Cat. sc-398394, Lot #G0571), STAT5 total (Cat. sc-74442, Lot #C1618) and HSC70 (Cat. sc-7298, Lot #E1914) (all Santa Cruz Biotechnology); phospho-STAT5 alpha Tyr694 (Cat. 71–6900, Lot #TF268386) (Thermo Fisher Scientific); Transferrin (Cat. ab82411; Abcam) and ApoB anti-serum (a gift from Dr. Larry Swift; Vanderbilt University Medical Center) all at a dilution of 1:1000 and secondary HRP-linked antibodies against mouse (Cat. 7076, Lot #32) and rabbit (Cat. 7074, Lot #26) (both Cell Signaling Technology) all at a dilution of 1:3000 or appropriate IRDye® secondary antibodies (LI-COR, Lincoln, NE) at a dilution of 1:15,000 were used. The blots were scanned with either the Fusion X (Vilber, France) or the LI-COR Odyssey (LI-COR, Lincoln, NE) and quantified with Image J (Version 1.48 v) or LI-COR ImageStudioLite (Version 5.2.5), respectively.

**Statistics**. The data are displayed as a mean ± SEM. Comparison among groups was performed by using ANOVA followed by repeated unpaired two-tailed-Student's t-tests. A $p$ value of <0.05 was considered statistically significant. In Supplementary Figures 2c, Supplementary Figure 2m the Pearson correlation coefficient was calculated with GraphPad Prism (Version 7).

**Reporting summary**. A Reporting Summary for this article is available as a Supplementary information file. Further information on research design is available in the Nature Research Reporting Summary linked to this article.

### Data availability

Uncropped scans of all blots presented in the main text are provided in Supplementary Figure 7. All the data supporting the findings of this study are available from the corresponding author on request.

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

## Acknowledgements

This study was supported by Austrian Science Fund (FWF) grant P26766 to T.S. and P30830 to C.F., the Medical Scientific Fund of the Mayor of the City of Vienna (Grant #15228) to T.S. and the NIH grants DK074873, DK083568, DK082724 to C.B. A.G. was supported by Horizon 2020 project EPoS-Elucidating Pathways of Steatohepatitis (under grant agreement no. 634413) and H2020-MSCA ITN-2016 FOIE GRAS (under grant agreement No 722619). The authors wish to thank the following individuals for their contributions: Gary J. Schwartz (Albert Einstein College of Medicine, NY), Anna Fenzl, Philipp Schwabl and Thomas Reiberger (all Medical University Vienna) for their input and technical support when establishing the liver sympathectomy, Ludger Scheja (University Medical Centre Hamburg - Eppendorf) and Herbert Stangl (Medical University Vienna) for discussing parts of the results section, Yiying Zhang and Streamson Chua, Jr (both Columbia University, NY and supported by P30 DK26687) for providing the LepRAPER mice, Larry Swift (Vanderbilt University Medical Center, TN) for donating the ApoB antibody, Juraj Krssak for his help with the liver lipid analyses, and Sameer Abu Eid for his help with some of the animal work. Further, we thank Thomas Metz (Medical University of Vienna) for his technical assistance in the acyl-carnitine analysis, the team of the Core Unit for Biomedical Research of the Medical University of Vienna, for animal care and the Preclinical Imaging Laboratory (PIL) of the Medical University of Vienna for their technical support with the MRS studies.

## Author contributions

Conceptualization, T.S., C.B. and M.T.H.; Methodology, T.S., C.F., C.B. and M.T.H.; Formal Analysis, Investigation, and Visualization, M.T.H., C.M.S., M.K., F.C., S.G., A.G., A.F., S.B., T.S., M.Z., C.F., T.H.H. and C.B.; Writing – Original Draft, M.T.H. and T.S.; Writing – Review and Editing, M.T.H., T.S., C.F., A.L. and C.B.; Supervision, Resources, and Funding Acquisition, T.S., C.F., and C.B.

## Additional information

**Competing interests:** The authors declare no competing interests.

