## [Peer Review File · Nature Communications]

Reviewers' comments:

Reviewer #1 (Remarks to the Author):

In this submission Hackl et al., describe an autonomic pathway linking brain activation by leptin to liver VLDL secretion and reduced de novo lipogenesis. The implication is that brain leptin action protects against the development of fatty liver disease, and the authors complete a compelling series of experiments that provide support for this assertion. First ICV leptin was shown to acutely increase VLDL secretion using the tyloxapol approach and chronic ICV infusion (28 days) decreased hepatic lipid content, while leptin receptor antagonist treatment increased liver lipid content. Careful control studies have been included including peripheral leptin infusion experiments and chemical-genetic deletion of peripheral leptin receptors. They go on to show that leptin induces an increase in hepatic MTP expression and reduces expression of genes involved in de novo lipogenesis. The CNS site of action of leptin was localized to the DVC using parenchymal injections. Finally, obesity is associated with both liver lipid accumulation and leptin resistance – the authors were able to demonstrate that high dose ICV leptin treatment is able to reduce liver lipid content in high-fat fed rats. NAFLD is a critical public health concern, and the studies highlight a potential therapeutic pathway to reduce NAFLD burden. The manuscript is well written.

The studies highlight a pathway that can, under experimental conditions (pharmacological or genetic gain- or loss-of function), lower liver lipid content, and that may have therapeutic relevance. The discussion around therapeutic potential, for example in lipodystrophy is enlightening. What is missing from the discussion is the role this pathway may play in “healthy” liver lipid homeostasis. Does liver lipid content vary consistently with changes in leptin tone over the course of the day/fasting/refeeding cycles? In other words, some discussion on the role/control strength of this pathway in the physiological range of leptin tone would be warranted.

The biggest weakness of the manuscript is the lack of CNS leptin signaling information in the high-fat feeding setting. For example, 28 days of high-fat feeding, or a total of 56 days is sufficient to induce leptin resistance (relative decrement in phosphoSTAT3 or other measure such as cfos) in many brain areas. These data suggest that the DVC is less susceptible to such an effect of high-fat feeding, OR that the dose of leptin was sufficiently high to overcome resistance. If brain tissues were available from a high-fat diet study cohort, data such as shown on Figure S1a for low-fat fed animals, would strengthen the paper.

How did leptin receptor antagonist infusion not influence food intake or body weight?

Are there known to be significant differences in parasympathetic innervation of liver in rodents relative to humans? If so, this would be important to recognize in the discussion.

Regarding therapeutic potential – could the authors consider with a little more granularity a clinical development pathway? Intranasal delivery of insulin and other peptides has been considered for many years, yet we currently do not clinically use this route.

Line 316 “of” should be deleted.

Reviewer #2 (Remarks to the Author):

This is an interesting study indicating that leptin signaling in the brain protects from hepatic steatosis by stimulating VLDL-TG secretion and reducing lipid production. Notably, these effects are mostly preserved in obese animals, if leptin is administered directly into the 3rd ventricle. The authors propose that stimulation of brain leptin signaling, e.g. by administration via an intranasal route, could be a strategy for amelioration of hepatic steatosis. The study is well planned and experimental designs appropriate.

1. The authors show apoB100 since this protein is exclusively synthesized in the liver. However, rat hepatocytes also produce apoB48. In fact, apoB48 secretion may exceed the apoB100

secretion. As it is feasible that the apoB mRNA editing is influenced by the different interventions, both apoB100 and apoB48 should be shown.

2. This reviewer does not fully understand the quantification of hepatic DNL. After labeling with heavy water, the contribution of DNL (measured as contribution of labelled palmitate) to VLDL-TG and liver-TG can be estimated. These data should be shown, before and after interventions, as the SCD1 index can be somewhat questionable.

3. The hepatic steatosis develops when the lipids influx exceeds outflux of the liver. DNL is closely linked to beta-oxidation as beta-oxidation is blocked by malonyl-Coa generated by DNL. It seems less likely that beta-oxidation is a key player, but it should be demonstrated directly.

4. As discussed, there are reports that leptin directly modulates lipid metabolism in isolated rat livers. It would be important to perform intervention in primary rat hepatocytes to clarify this ambiguity.

We thank the reviewers for their careful reading of our manuscript and the helpful comments and advice they have given us. We feel that based on the reviewer's comments we substantially strengthened the manuscript. We now include additional data sets on a) FAS activity in vagotomized animals (Supplementary Figure 4m); b) circulating apoB48 levels as requested (Figs. 1d,e and I,j); c) brain leptin signaling data in HFD fed rats as requested (Supplementary Figure 5d, e), d) IP leptin signaling data in HFD fed rats (Supplementary Figure f, g), e) correlation of liver and plasma TAG enrichments (Supplementary Figure 2m) and f) acyl-carnitines in plasma (Supplementary Figure 6c and d). Furthermore, we improved and extended the discussion by including the points raised by the reviewers.

Note that all changes to the previous manuscript are marked in green.

Reviewer #1 (Remarks to the Author):

In this submission Hackl et al., describe an autonomic pathway linking brain activation by leptin to liver VLDL secretion and reduced de novo lipogenesis. The implication is that brain leptin action protects against the development of fatty liver disease, and the authors complete a compelling series of experiments that provide support for this assertion. First ICV leptin was shown to acutely increase VLDL secretion using the tyloxapol approach and chronic ICV infusion (28 days) decreased hepatic lipid content, while leptin receptor antagonist treatment increased liver lipid content. Careful control studies have been included including peripheral leptin infusion experiments and chemical-genetic deletion of peripheral leptin receptors. They go on to show that leptin induces an increase in hepatic MTP expression and reduces expression of genes involved in de novo lipogenesis. The CNS site of action of leptin was localized to the DVC using parenchymal injections. Finally, obesity is associated with both liver lipid accumulation and leptin resistance – the authors were able to demonstrate that high dose ICV leptin treatment is able to reduce liver lipid content in high-fat fed rats. NAFLD is a critical public health concern, and the studies highlight a potential therapeutic pathway to reduce NAFLD burden. The manuscript is well written.

The studies highlight a pathway that can, under experimental conditions (pharmacological or genetic gain- or loss-of function), lower liver lipid content, and that may have therapeutic relevance. The discussion around therapeutic potential, for example in lipodystrophy is enlightening.

We thank the reviewer for her/his encouraging remarks.

What is missing from the discussion is the role this pathway may play in “healthy” liver lipid homeostasis. Does liver lipid content vary consistently with changes in leptin tone over the course of the day/fasting/refeeding cycles? In other words, some discussion on the role/control strength of this pathway in the physiological range of leptin tone would be warranted.

This is an interesting question. Although leptin secretion in humans is moderately modulated by feeding (+16%) and fasting (-30%) as well as circadian rhythmicity (18% amplitude of the 24h mean)(1, 2), short-term postprandial changes in circulating leptin do not necessarily regulate satiety (3), a major function of leptin action. Thus, we believe that leptin, which is chiefly secreted in proportion to body fat stores, regulates energy metabolism rather long-term and may not play a critical role in the minute to minute control of metabolism such as in the fasting to feeding transition. In keeping with this notion, the acute brain leptin infusion did not change hepatic lipid content in our study. A role of brain leptin became only apparent after chronic leptin infusion over the course of 2-4 weeks. We now briefly discuss this in our revised manuscript.

Since we found that only a chronic leptin infusion into the cerebroventricular space led to measurable changes in hepatic lipid content, we believe that leptin, which is chiefly secreted in proportion to body fat stores, regulates hepatic lipid content rather long-term than in the minute to minute control of metabolism such as in the fasting to feeding transition. This notion is supported by the findings that leptin levels are only moderately modulated in humans by feeding, fasting or circadian rhythmicity^{51,52}.

The biggest weakness of the manuscript is the lack of CNS leptin signaling information in the high-fat feeding setting. For example, 28 days of high-fat feeding, or a total of 56 days is sufficient to induce leptin resistance (relative decrement in phosphoSTAT3 or other measure such as cfos) in many brain areas. These data suggest that the DVC is less susceptible to such an effect of high-fat feeding, OR that the dose of leptin was sufficiently high to overcome resistance. If brain tissues were available from a high-fat diet study cohort, data such as shown on Figure S1a for low-fat fed animals, would strengthen the paper.

The reviewer raises an excellent point. As requested we measured STAT3 phosphorylation in biopsies of the MBH and the DVC harvested after the chronic ICV leptin/vehicle infusion experiments and indeed find intact leptin signaling in the DVC when leptin is given directly into the cerebroventricular space. However, in keeping with a transporter defect of leptin across the blood brain barrier IP leptin injection fails to induce STAT3 phosphorylation in HFD fed rats in the DVC and the MBH, respectively (Supplementary Figure 5d,e and 5 f,g).

How did leptin receptor antagonist infusion not influence food intake or body weight?

We believe that the reason for the lack of changes in food intake in our study is due to the low dose we used. Compared to others (4) we used a much lower dose of the leptin receptor antagonist. We mention this now in the revised manuscript. We chose such a low dose intentionally as we aimed to avoid an effect of the antagonist on food intake that otherwise would have confounded our study.

“The lack of orexigenic action of the LpR antagonist is likely due to the fact that we used a much lower dose of the LpR antagonist compared to other studies³⁶.”

Are there known to be significant differences in parasympathetic innervation of liver in rodents relative to humans? If so, this would be important to recognize in the discussion.

Excellent question. While there are considerable differences in sympathetic innervation patterns of the liver between rats and humans (portal area only vs. portal area plus sinusoids), cholinergic nerve fiber distribution shows similar patterns between both species (portal area only) (5, 6). We discuss this now briefly in the manuscript.

“In this context it is important to point out that, while there are considerable differences in sympathetic innervation patterns of the liver between rats and humans, cholinergic nerve fiber distribution shows very similar patterns in both species.”

Regarding therapeutic potential – could the authors consider with a little more granularity a clinical development pathway? Intranasal delivery of insulin and other peptides has been considered for many years, yet we currently do not clinically use this route.

The reviewer raises a very valid point. While intranasal leptin administration shows promising results in rodent studies a clinical application in humans has limitations: Relatively high doses have to be applied when peptides are administered intranasally. Since recombinant leptin is expensive a human trial with intranasal leptin is currently not feasible. In addition, variable absorption via the nasal mucosa may also be an issue. However, possible approaches to circumvent some of these drawbacks could be to a) engineer leptin analogs that have improved blood brain barrier permeability features (7, 8) or b) combine leptin analogs with drugs that restore leptin sensitivity (9). We discuss this now in more detail.

“Based on these findings it seems promising to test if intranasal leptin overcomes leptin resistance in patients with obesity and type 2 diabetes. However, while intranasal leptin administration shows promising results in rodent studies a clinical application in humans has certain limitations: Relatively high doses have to be applied when peptide hormones are administered intranasally. Since recombinant leptin is expensive a human trial with intranasal leptin is currently not viable. In addition, variable absorption via the nasal mucosa may be an issue. However, possible approaches to circumvent some of these drawbacks could be to engineer leptin analogs with improved BBB permeability features^{65,66} or to co-infuse leptin with peptides or drugs that restore leptin sensitivity in the obese state⁶⁷.”

Line 316 “of” should be deleted.

Thanks for pointing this out.

Reviewer #2 (Remarks to the Author):

This is an interesting study indicating that leptin signaling in the brain protects from hepatic steatosis by stimulating VLDL-TG secretion and reducing lipid production. Notably, these effects are mostly preserved in obese animals, if leptin is administered directly into the 3rd ventricle. The authors propose that stimulation of brain leptin signaling, e.g. by administration via an intranasal route, could be a strategy for amelioration of hepatic steatosis. The study is well planned and experimental designs appropriate.

We thank the reviewer for her/his positive remarks.

1. The authors show apoB100 since this protein is exclusively synthesized in the liver. However, rat hepatocytes also produce apoB48. In fact, apoB48 secretion may exceed the apoB100 secretion. As it is feasible that the apoB mRNA editing is influenced by the different interventions, both apoB100 and apoB48 should be shown.

We agree with the reviewer. We now show both apoB100 and apoB48 in the revised manuscript (see revised Figs. 1d,e and 1i,j).

2. This reviewer does not fully understand the quantification of hepatic DNL. After labeling with heavy water, the contribution of DNL (measured as contribution of labelled palmitate) to VLDL-TG and liver-TG can be estimated. These data should be shown, before and after interventions, as the SCD1 index can be somewhat questionable.

We are sorry that we did not sufficiently outline our methodological approach here. We like to emphasize that we do not only measure palmitate but instead analyzed the % enrichment in each distinct triglyceride class to arrive at a more comprehensive assessment of hepatic de novo lipogenesis. By only looking at palmitate, which can, for example, be elongated into stearate during de novo synthesis, one may underestimate total hepatic de novo lipogenesis. We explain this more clearly now in the method part and also improved the x-axis labeling for more clarity in Figure 3j.

In terms of showing data before and after an intervention, this is tricky with D2O as a stable isotope tracer. The clearance of D2O is very long and takes weeks, therefore short-term repetitions of experiments are not possible. Since we are also looking at liver tissue directly a liver biopsy before cannulation and minipump implantation would have been necessary to assess baseline conditions in the same animal. This further complicates the experiment and is problematic from an animal ethics standpoint. We therefore chose to use a weight- and age-matched control group that received vehicle to assess the effects of brain leptin on hepatic de novo lipogenesis.

In addition, we now added data on plasma TAG enrichments, which reflect our liver data in correlation analyses (see Supplementary Figure 2m).

3. The hepatic steatosis develops when the lipids influx exceeds outflux of the liver. DNL is closely linked to beta-oxidation as beta-oxidation is blocked by malonyl-Coa generated by DNL. It seems less likely that beta-oxidation is a key player, but it should be demonstrated directly.

We agree with the reviewer, based on the fact that ketone bodies are decreased and hepatic PPAR alpha expression is not changed, hepatic beta-oxidation is unlikely to play an important role in our experimental setup. In addition to this, we now also added long-chain acyl-carnitine measurements in plasma showing no major differences in concentrations of long-chain acyl-carnitines between chronic/acute brain leptin and controls (Supplementary Figure 6c and d). These lipid-derived long-chain acyl-carnitines are known to be increased in situations where beta-oxidation (i.e. HFD feeding, fasting) is stimulated (10). We feel that we now have sufficient evidence that hepatic beta-oxidation is not a key player that explains the changes in hepatic lipid content we observe.

4. As discussed, there are reports that leptin directly modulates lipid metabolism in isolated rat livers. It would be important to perform intervention in primary rat hepatocytes to clarify this ambiguity.

While we agree with the reviewer that this question deserves clarification in general, we want to point out that the characterization of possible direct effects of leptin on the hepatocyte was not the scope of this paper. We set out to assess leptin's brain effects on hepatic lipid metabolism and performed several control experiments that rule out that peripheral leptin signaling plays a major role in our experimental setup.

References:

1. J. Dallongeville *et al.*, Short term response of circulating leptin to feeding and fasting in man: influence of circadian cycle. *Int J Obes Relat Metab Disord* **22**, 728-733 (1998).
2. C. Simon, C. Gronfier, J. L. Schlienger, G. Brandenberger, Circadian and ultradian variations of leptin in normal man under continuous enteral nutrition: relationship to sleep and body temperature. *J Clin Endocrinol Metab* **83**, 1893-1899 (1998).
3. M. Romon *et al.*, Leptin response to carbohydrate or fat meal and association with subsequent satiety and energy intake. *Am J Physiol* **277**, E855-861 (1999).
4. J. Zhang, M. K. Matheny, N. Tumer, M. K. Mitchell, P. J. Scarpace, Leptin antagonist reveals that the normalization of caloric intake and the thermic effect of food after high-fat feeding are leptin dependent. *Am J Physiol Regul Integr Comp Physiol* **292**, R868-874 (2007).
5. H. Akiyoshi, T. Gonda, T. Terada, A comparative histochemical and immunohistochemical study of aminergic, cholinergic and peptidergic innervation in rat, hamster, guinea pig, dog and human livers. *Liver* **18**, 352-359 (1998).
6. K. J. Jensen, G. Alpini, S. Glaser, Hepatic nervous system and neurobiology of the liver. *Compr Physiol* **3**, 655-665 (2013).
7. X. Yi *et al.*, Pluronic modified leptin with increased systemic circulation, brain uptake and efficacy for treatment of obesity. *J Control Release* **191**, 34-46 (2014).
8. T. O. Price *et al.*, Transport across the blood-brain barrier of pluronic leptin. *J Pharmacol Exp Ther* **333**, 253-263 (2010).
9. C. Clemmensen *et al.*, GLP-1/glucagon coagonism restores leptin responsiveness in obese mice chronically maintained on an obesogenic diet. *Diabetes* **63**, 1422-1427 (2014).
10. T. R. Koves *et al.*, Mitochondrial overload and incomplete fatty acid oxidation contribute to skeletal muscle insulin resistance. *Cell Metab* **7**, 45-56 (2008).

REVIEWERS' COMMENTS:

Reviewer #1 (Remarks to the Author):

The authors have been responsive to the review.

Reviewer #2 (Remarks to the Author):

This reviewer is not fully convinced, but all the revisions are fully OK.

Point-by-point Response to Reviewers:

REVIEWERS' COMMENTS:

Reviewer #1 (Remarks to the Author):

The authors have been responsive to the review.

We again thank the reviewer for her/his comments to improve our manuscript.

Reviewer #2 (Remarks to the Author):

This reviewer is not fully convinced, but all the revisions are fully OK.

Skepticism is an important prerequisite of science. We thank the reviewer for her/his insights.